# The Baltimore Urban Food Distribution (BUD) App: Study Protocol to Assess the Feasibility of a Food Systems Intervention

**DOI:** 10.3390/ijerph19159138

**Published:** 2022-07-26

**Authors:** Joel Gittelsohn, Emma C. Lewis, Nina M. Martin, Siyao Zhu, Lisa Poirier, Ellen J. I. Van Dongen, Alexandra Ross, Samantha M. Sundermeir, Alain B. Labrique, Melissa M. Reznar, Takeru Igusa, Antonio J. Trujillo

**Affiliations:** 1Human Nutrition, Department of International Health, Johns Hopkins Bloomberg School of Public Health, Baltimore, MD 21205, USA; elewis40@jhu.edu (E.C.L.); nmarti38@jhu.edu (N.M.M.); lpoirie4@jhmi.edu (L.P.); srex2@jhmi.edu (S.M.S.); 2Department of Civil and Systems Engineering, Johns Hopkins Whiting School of Engineering, Baltimore, MD 21205, USA; szhu34@jhu.edu (S.Z.); tigusa1@jhu.edu (T.I.); 3GGD Noord-en Oost-Gelderland, 7231 AC Warnsveld, The Netherlands; ellen.vdongen@gmail.com; 4Nutrition Epidemiology, School of Global Public Health, University of North Carolina Gillings, Chapel Hill, NC 27599, USA; alexandraa.ross@outlook.com; 5Global Disease Epidemiology and Control, Department of International Health, Johns Hopkins Bloomberg School of Public Health, Baltimore, MD 21205, USA; alabriq1@jhu.edu; 6Interdisciplinary Health Sciences, Oakland University School of Health Sciences, Detroit, MI 48309, USA; reznar@oakland.edu; 7Health Systems, Department of International Health, Johns Hopkins Bloomberg School of Public Health, Baltimore, MD 21205, USA; atrujil1@jhu.edu

**Keywords:** food access, retailers, obesity, urban, multilevel interventions, policy, study design

## Abstract

Low-income urban communities in the United States commonly lack ready access to healthy foods. This is due in part to a food distribution system that favors the provision of high-fat, high-sugar, high-sodium processed foods to small retail food stores, and impedes their healthier alternatives, such as fresh produce. The Baltimore Urban food Distribution (BUD) study is a multilevel, multicomponent systems intervention that aims to improve healthy food access in low-income neighborhoods of Baltimore, Maryland. The primary intervention is the BUD application (app), which uses the power of collective purchasing and delivery to affordably move foods from local producers and wholesalers to the city’s many corner stores. We will implement the BUD app in a sample of 38 corner stores, randomized to intervention and comparison. Extensive evaluation will be conducted at each level of the intervention to assess overall feasibility and effectiveness via mixed methods, including app usage data, and process and impact measures on suppliers, corner stores, and consumers. BUD represents one of the first attempts to implement an intervention that engages multiple levels of a local food system. We anticipate that the app will provide a financially viable alternative for Baltimore corner stores to increase their stocking and sales of healthier foods, subsequently increasing healthy food access and improving diet-related health outcomes for under-resourced consumers. The design of the intervention and the evaluation plan of the BUD project are documented here, including future steps for scale-up. Trial registration #: NCT05010018.

## 1. Introduction

Low-income communities in the United States are characterized by low access to healthier food and consequent food insecurity and diet-related chronic disease [1]. Patterns of food access are related to food source types present, including low access to supermarkets and grocery stores, and high access to small convenience/corner stores, fast food, and carryout restaurants [2]. Food access has been defined in terms of availability, cost, quality, and location [3]. Low-income communities tend to be weak in all of these dimensions, including having low availability of healthier foods, high prices, low quality, and poor location (e.g., within store, not well displayed). Over the years, evidence has accumulated that the type of nearby food sources is related to consumers’ diet and their health, with access to nearby supermarkets and grocery stores reducing risk for diet-related chronic disease, and small convenience stores being associated with poor diet and negative health outcomes [4]. Baltimore, Maryland is an example of this challenge, with numerous reports documenting low healthy food access [5], high food insecurity [6], and high rates of diet-related chronic disease [7].

Most previous works on food access have looked at these dimensions from the perspective of community members (consumers) [8,9,10,11,12,13,14]. The composition of the local retail food environment plays an important role in shaping community food acquisition patterns and food security. To this end, many community-based interventions seeking to improve food access have largely targeted community members by working in food sources to increase access at local food sources, including work in small food stores [15] and prepared-food sources [16]. Despite these substantial efforts, limited work has been conducted to improve access to healthier foods for the food sources themselves.

Why is food sourcing for food stores themselves a concern? Low access to healthy food in Baltimore and similar urban food systems is largely due to a distribution gap between small independent food sources and larger wholesalers/distributers. Among small urban corner stores, unhealthy foods and beverages are much more accessible to these stores due to incentive programs, convenient and often free delivery services, and other formal and informal incentives provided by suppliers [17]. On the other hand, these same small urban retailers have limited access to healthier foods due to the lack of such programs offered by fresh food suppliers [12].

The intervention described in this paper is a true food systems intervention. Food systems are complex networks of relationships between different types of actors, including producers, manufacturers, suppliers/distributors, retailers, and consumers [18]. In our view, a food systems intervention would target multiple levels of actors within a food system (e.g., producers, distributors, retailers, consumers) simultaneously, and focus on improving the interactions between these levels. Therefore, one approach to addressing the problem of healthy food sourcing/distribution is to implement multilevel food system interventions that can sustainably improve healthy food access for small food retailers—especially in low resource settings.

Digital applications have been used previously to modify food access [19]. During the coronavirus (COVID-19) pandemic, use of online shopping services doubled [20]. However, the benefits of these services have not been equally distributed. For example, during the pandemic, online shopping services were not available for WIC participants, and only available in some stores for SNAP recipients. It is crucial to improve access to healthier foods among our most vulnerable populations. Digital interventions have the potential to greatly improve the reach of food access interventions, while keeping down costs.

The overall goal of the Baltimore Urban food Distribution (BUD) trial is to develop a digital application (app) to improve access to healthier foods and beverages in small corner stores in low-income urban settings and to generate preliminary findings in support of a full-scale clinical trial. The BUD app will improve access to healthier foods by linking local suppliers (producers, wholesalers) with small store owners, and provide cost-effective options to order and deliver healthier foods. These options will include collective purchasing and shared delivery options, which will reduce costs and enhance sustainability.

The BUD trial has two primary aims:1.To conduct formative research and engage key stakeholders to develop a user-friendly BUD app;2.To implement a randomized controlled trial pilot study of the BUD app and assess its feasibility;

Then there is one secondary aim:3.To demonstrate the impact of BUD in terms of small urban food store stocking and sales of healthier foods.

## 2. Methods/Design

### 2.1. BUD Trial Study Design Overview

The BUD study is divided into three main phases. First, we are conducting formative research in Baltimore City to inform the design of a user-friendly interface and experience for corner store owners and their suppliers (local producers, wholesalers), as well as develop a stable version of the app. Second, we are employing a randomized controlled trial study design over 8 months to pilot-test the app, introducing different features in stages to identify and resolve challenges, and assessing feasibility and preliminary efficacy in terms of change at the store level. Finally, we will show the final app to key stakeholders (corner store owners, suppliers) in other cities to assess transferability and potential for dissemination.

### 2.2. Setting

The BUD study takes place in East Baltimore, composed of predominantly African American communities, which represents about one-third of the city’s area. A high proportion of the households in East Baltimore are low income. Baltimore City as a whole has 633 corner stores, 185 convenience stores, 47 supermarkets, 18 farmers markets, 24 urban farm sites, and 6 public markets [21]. In East Baltimore, there are 12 supermarkets, approximately 168 corner stores, 88 convenience stores, 2 farmers markets, 9 urban farms, and 1 public market. Of the corner stores, about two-thirds are owned and operated by Korean Americans, roughly 15% are operated by Chinese Americans, 5% by Hispanic Americans, with smaller fractions owned by persons of different sociocultural backgrounds.

Most corner store owners locate their stores in areas not being served by supermarkets. Baltimore corner stores typically stock few affordable healthy foods and beverages, instead stocking higher-priced nutrient-poor foods and beverages. Corner stores have significantly lower healthy food availability index (HFAI) scores than those located in predominantly middle- to high-income areas [5,21,22]. In fact, in 2018, a sample of 525 Baltimore corner stores scored an average HFAI score of 9.1 out of 28.5, which is considered to be low (the average supermarket HFAI score in this setting was 27.7) [5].

### 2.3. Inclusion and Exclusion Criteria

The trial includes 38 small corner stores located in East Baltimore randomized into treatment (*n* = 19) or comparison (*n* = 19) groups. Eligible corner stores are located in areas identified as a Healthy Food Priority Area [5], are located >0.25 miles from a supermarket, and are classified as a small store (≤ four aisles, ≤ two cash registers). In addition, the store owner/manager should be English, Korean, Spanish, or Mandarin speaking as a first language, willing and able to order food through a smartphone or internet-enabled device, and willing to attend in-store trainings on the use of the BUD app.

Wholesalers and producers are eligible to participate who currently serve Baltimore City (e.g., have wholesale locations and/or attend farmers markets in the Baltimore area) and are willing to use the BUD app.

Consumers will be identified by the store owners as a regular customer of the store (i.e., purchase food items at least once a week in the store) and will be 21–75 years old, live/work within a 0.5-mile radius from one of the participating corner stores, and live in a household of at least two persons. Consumers will be excluded from participation when they anticipate moving out of Baltimore City in the next 12 months or when they are pregnant, as we are assessing change in weight as a secondary outcome.

### 2.4. Randomization of Corner Stores to Treatment

Corner stores will be randomized into intervention (receiving the BUD app) or comparison (not receiving the BUD app) groups, following completion of the baseline assessments. The names of each of the 38 corner stores will be written on a separate piece of paper, mixed up, chosen out of a bowl by community members, and alternately assigned to intervention or comparison status. Randomization will be performed publicly and documented via photo and posted on team social media sites. We found that public randomization conducted this way serves to reduce community concerns with bias in the selection process. Participating stores and interventionists will not be blind to the intervention status due to the nature of the intervention design.

### 2.5. Formative Research

Extensive multistage formative research has and will be conducted to aid in intervention planning and app development.

Completed formative research: Prior to receiving funding, we conducted initial formative research on improving food access for corner stores and the acceptability of an app to support this goal [23]. Interviews with corner store owners, wholesalers, and food environment experts were conducted to understand the context of food distribution and supply in the city and to identify features that would make the BUD app acceptable, operable, useful, and user-friendly. A visual mockup, or wireframe, of the BUD app was developed and presented to store owners to stimulate additional feedback. Some of the key findings were: (1) the BUD app needs to provide more than just fresh produce to be considered useful by small store owners, and (2) delivery will be challenging and will require multiple options.

Additional formative research: As part of the formative phase of the current study, we are conducting additional formative research to aid in app design. This includes: (1) conducting case studies of 10 different corner stores and carefully examining their stocking, ordering, and pricing decision making/strategies, methods of payment, and so on; (2) interviewing 20 community members to assess their priorities for food stocking in local small stores; and (3) developing a prototype of the BUD app that mimics the process of using a completed app—intended to stimulate further discussion of strengths and challenges. Some of the key questions we are trying to answer include: (1) how best to frame the BUD app in a way that encourages cooperation and use of a collective purchasing feature, (2) what payment options are most acceptable, (3) what delivery options will be most acceptable and affordable, (4) what additional features are necessary and/or desirable (e.g., some way to assess and report on consumer demand for specific products), and (5) how we design a user-friendly interface and experience for corner store owners.

### 2.6. App Development Process

The development of a user-friendly and functional web-based smartphone app depends on the thoughtful design and development of both the “front-end” and “back-end” elements of the app. The front-end includes the user interface or the screens where the user interacts with the software. The back end consists of the server that provides data on request, the app that channels it, and the database that organizes the information. There are three types of users for the BUD app, which are suppliers, retailers, and administrators. Administrators will be study team members and have the ability to monitor the usage of the app and award BUDCredit, among other functions. Two versions of the BUD app will be designed: the first will be for suppliers and retailers to use on mobile devices, and the second will be a desktop/laptop-aimed admin page designed for administrators to review analytic results and monitor the BUD app data. The research study team will have access to the BUD app usage data through the admin page. They can check for any communication issues and get feedback on app problems. Additionally, the team can monitor prices and check overall analytics results through the admin page.

#### 2.6.1. User Interface and Experience Design Process

To ensure high usability of the BUD app, this interface will be designed and tested through formative research in partnership with target users, as elucidated in the following steps. First, we will design the flow of information and map out the intended experiences we would like users to have while navigating the BUD app. Based on the formative research and user personas, we will design the preliminary structure of how information, content, and features are organized and arranged within the app. We will visualize this organization via a sitemap, a hierarchical diagram showing the structure of content and intended experiences within a website or app. Next, via three qualitative research methods (tree testing, card sorting, and in-depth interviews), we will assess the logic of the sitemap and the findability of content within the hierarchy of the tool.

Once the map has been tested and revised, we will then generate initial mockups, called low-fidelity wireframes, of the user interface. These show the first sketches of how the interface will be designed. Using these low-fidelity wireframes, we will repeat the tree testing (a method to measure the findability of content and tools) with potential users. Once we are confident that the design is passing these initial user performance tests, we will then generate and test high-fidelity wireframes (mockups that include all design elements and branding) with target users. The sitemap, branding, and low- and high-fidelity mockups are designed via Adobe Photoshop, Illustrator, and XD. Subsequently, these high-fidelity wireframes will be programmed into an app prototype via Adobe XD. A prototype is a clickable visualization of the BUD app that permits likeability, findability, and usability testing with target users. Tests will include running two to three use cases with potential users where participants are given a typical task to complete using the app. The time to complete a task and slow points are documented. Finally, we will convert the user interface from the graphic design of the prototype into an open-access package of code (HTML, CSS, and JavaScript) and content that will be uploaded to an open-access code repository and shared with the back-end developer team.

#### 2.6.2. Development of Back-End Elements

The back end of the BUD app will need to be able to communicate with users in real time and store, update, and track both purchases and deals made through the app. Additionally, it will need to securely collect and store private, sometimes sensitive, data, such as in-app usage statistics (e.g., number of deals, purchases, chats, delivery methods, payment methods). We will also use the BUD app to collect information on the app’s usability and likeability, frequency of deals and purchases, and user satisfaction. Finally, we will need to be able to make iterative updates to the app, as user feedback is collected.

To address these needs, we will host the BUD app on Google Firebase. Firebase is an online platform or tool that streamlines the BUD app development process and provides centralized access to, for example, analytics, authentication, databases, configuration, file storage, and push messaging. Another strength of Firebase is that all services are hosted in the cloud, making collaboration and communication across the team easier. A specific URL for the BUD app will be purchased, and users will be able to access the BUD app through the link.

### 2.7. Usability Testing and Prepilot

Once we have a functioning app, we will ensure user-friendliness and functionality through a series of usability tests, culminating in a community “prepilot”. Initial usability tests will involve providing the BUD app to 8–10 target users (local producers, wholesalers, corner store owners) and asking them to complete certain tasks (e.g., set up a collective purchasing deal, order a food, etc.). Subsequently, one small urban producer and two to three small corner stores located in East Baltimore will participate in a prepilot usability test of BUD prior to full-scale implementation of the pilot trial. The research team will elicit user feedback on acceptability, operability, and perceived sustainability from both the producer and store owner perspectives.

### 2.8. Participants and Evaluation Sample Recruitment

Informed consent will be obtained for the evaluation sample by Collaborative Institutional Training Initiative-trained and certified data collectors. Description of information collected is provided in the Evaluation Measures section.

Recruitment of suppliers (producers, wholesalers, and so on): To facilitate recruitment of food suppliers, including, but not limited to, local farmers, wholesalers, and other produce distributors, we will have staff speak with eligible owners and managers asking them to participate in the intervention. The team will create information sheets regarding expectations for participation, such as use of the BUD app, information on delivery methods, and the amount of product required to be uploaded for sale. Recruitment will take place at local farmers markets, urban farms, and other community settings, and via word of mouth.

Recruitment of corner stores: Thirty-eight corner stores located in East Baltimore will be recruited. To facilitate recruitment, we will have bilingual project staff approach eligible store owners. We will distribute flyers with answers to frequently asked questions (FAQ) about the trial written in English, Korean, Spanish, and Mandarin (the most common languages spoken by corner store owners in Baltimore City). Consent forms will be prepared in the four languages, and will clearly explain the benefits and potential risks of participating in the trial, including detailed description of the BUD app, promotional materials, and small initial subsidy (BUDCredit), which will cover some of the purchasing/delivery costs for using the BUD app. We will document reasons for participation and refusal to participate in the study.

Recruitment of consumers: Our sample of consumers will allow us to describe regular customers of participating corners stores and to assess preliminary efficacy of the BUD app at the consumer level. Five consumers will be recruited at each participating study store (*n* = 190) and will be referred by the store owner or staff as a “usual” shopper (i.e., purchases food items from the store at least once a week). This will also improve the likelihood of retention in the study. Furthermore, one of the eligibility criteria of the consumer sample is to not anticipate moving out of Baltimore City for one year. We will conduct interviews at the nearby Johns Hopkins Bloomberg School of Public Health or in other community locations (e.g., recreation centers), following recruitment. A gift card for participation will also facilitate recruitment. We will document all refusals, including the reason for refusal.

### 2.9. Intervention Description

The BUD food systems intervention is depicted in Figure 1. The BUD intervention includes three primary components: (1) the BUD app itself, (2) training of store owners and suppliers in the use of the app, and (3) the provision of promotional materials for use at the point of purchase. *BUD App*: The BUD app will have three main modules: supplier, retailer, and administrator. The key features of the supplier module include registration, entering products onto the BUD app for sale, setting BuddyUp deals, providing delivery options and costs, and tracking and filling orders (Figure 2). BuddyUp deals use the power of collective purchasing; suppliers can offer a reduced price for selected products that are purchased in bulk by multiple small store owners. The key features of the retailer (corner store) module include registration, identification of products for purchase, review of available BuddyUp deals, a multilanguage chat feature that allows store owners to coordinate group purchases, delivery option selection (including BuddyLift), and options for payment and tracking (Figure 3). BuddyLift is a delivery option that permits one member of a BuddyUp deal to pick up and deliver the foods to the other members, in exchange for a reduced price. The administrator module is a portal for tracking app usage statistics and for generating and analyzing user satisfaction data.

Training and Stages of Implementation: The BUD pilot will be implemented in four 2-month stages, characterized by the gradual introduction of app features, the incremental introduction of food products, the presence of an initial subsidy to overcome resistance among store owners in phases 1 and 2 only, and the promotion of targeted foods and beverages within corner stores (Table 1). In the first 2 weeks of each stage of implementation of the pilot, in-person training in the use of the new features will be provided to the intervention stores. The first phase of the pilot will involve the introduction of the BUD app’s general features and focus on the use of the app to order and receive beverages. The second pilot phase will introduce the BuddyUp feature and introduce fresh produce. The third phase will introduce the BuddyLift delivery feature and incorporate whole grain products. The final phase will center on enhancing the use of all features of the BUD app.

Promotion of Healthy Foods: While the BUD app will support the provision of all foods and beverages to corner stores, special emphasis will be provided on healthier products. We will reach out to local producers of fruits and vegetables and especially encourage their participation. An incentive system for purchasing healthy foods will be offered to participating corner stores, leading to additional deals, reducing the costs. Initial subsidies for BuddyUp deals will only be offered for healthier products. Point-of-purchase in-store promotional materials (shelf talkers, posters) will be provided to stores to build awareness and generate interest in newly stocked healthier products.

### 2.10. Comparison Group

The 19 comparison stores will be introduced to the BUD app following completion of postintervention data collection.

### 2.11. Evaluation Measures

The BUD pilot trial will be evaluated pre- and postintervention at the supplier (producer, wholesaler), retailer (corner store), and consumer levels (Table 2). Differences in changes between the intervention and comparison samples will allow us to determine the impact of the intervention. Process evaluation measures will assess the reach, dose delivered, and fidelity of intervention implementation according to set standards. Participant retention and follow-up will be promoted by regular communication with producers, wholesalers, and stores. This will be in the form of in-person visits to stores, communication via the app/telephone calls, and trying to resolve any issues that have led to a need or indication to drop out of the study. All measures will be performed in intervention and comparison store and consumer samples, with the exception of process measures, which will be completed in the intervention stores only.

Supplier measures: Participating producers and wholesalers will be assessed pre-, post-, and during implementation of the pilot. Impacts will examine indicators, such as number of (1) orders from corner stores received using the app, (2) BuddyUp deals initiated, and (3) units of promoted foods and beverages sold to/delivered to corner stores. Process measures will include any use of the BUD app (reach), frequency and duration of use (dose), number of orders fulfilled, and number of BuddyUp deals offered (fidelity).

Retailer measures: Impact data for corner stores will be collected using a modified version of our Store Impact Questionnaire (SIQ) to assess store characteristics, self-reported BUD app usage, corner store owner psychosocial characteristics (self-efficacy, intentions, expectations), sales (last 7 days), and prices of promoted foods and beverages. In addition, we will introduce a point-of-sale (POS) tablet with software to record on-the-go unit sales of targeted foods and beverages [24]. Retailer process measures will center on the number of stores using the BUD app (reach), which features are used (dose) and how often the BUD app is used, as well as what types of functions they use in the app (i.e., BuddyLift or chat feature) (feasibility) and number of orders placed.

Consumer measures: We will assess consumer outcomes using a modified version of our Adult Impact Questionnaire (AIQ) that we used in previous studies in Baltimore [25]. This instrument will collect sociodemographic information (age, sex, education level, income, etc.) and food purchasing behavior, such as how many times a product was bought and from which type of store (i.e., corner store, grocery store, urban farm, etc.). In particular, we will assess the frequency of purchasing of promoted products from participating corner stores. The AIQ also requires data collectors to measure the height, weight, and body fat percentage of consumer participants. The Adult Block Food Frequency Questionnaire (FFQ) will estimate adult consumer food intake and nutrient consumption (e.g., total energy intake, total fat, added sugar, sugar-sweetened beverage, and fruit and vegetable intake) [26].

Process evaluation measures: We will record the delivery of the intervention by study team members during the intervention phase. Reach measures include the number of visits or consultations with retailers or suppliers per week. Dose standards are based on the amount of time spent interacting with retailers and suppliers, and fidelity standards capture the opinions on BUD from retailers and suppliers. Team process evaluation measures will be recorded using a process evaluation data form created via REDCap and collected on a tablet during a site visit [27,28].

Recruitment and retention: We will assess the recruitment and retention of suppliers, retailers, and consumers in the study using recruitment sheets used by trained interview staff during the recruitment phase of activities, as well as tracking sheets to capture the retention of participants during the implementation of the intervention.

### 2.12. Data Management

All collected data will be reviewed by two different trained research assistants for completeness. For data collected via paper forms, research assistants will de-identify and enter data into password-protected Microsoft Access databases.

### 2.13. Sample Size and Power

For the primary aims of this feasibility study, our study sample is based on the entire group of corner stores and suppliers who meet the inclusion criteria and agree to participate in the trial. A sample size calculation was conducted to assess the impact on our secondary aim: the impact on corner stores. The calculation was based on the analysis of a simple difference in change in mean HFAI scores. We used the reported mean change in HFAI in control and intervention corner stores from the previous B’more Healthy Communities for Kids trial [29], positing that the subsidy and price incentives combined with the introduction of the BUD app would create a greater change in HFAI than in the BHCK trial. Ultimately, calculations for a two-sample comparison of means assuming a difference in HFAI of 6 points, variance of 24, intraclass correlation of 0.0001, type I error of 5%, and power level set at 80% led to a total sample size of 34 stores with 17 per arm. Two stores were added to each arm of the study to account for possible attrition, leading to the final sample size of 38 stores.

This trial is not powered to show impact at the consumer level, but is designed to provide adequate precision of estimates to power a future clinical trial at the consumer level. By examining correlations of outcomes within stores over time, between participants (customers) within stores, and within participants over time, these data will allow us to assess clustering, permitting us to estimate the intraclass correlation coefficient (ICC) and precision around it. The consumer sample size is intended to obtain an estimate of effect size and the variance in food consumption, purchasing, and body mass index in order to plan a larger clinical trial.

### 2.14. Data Analysis and Statistical Analysis

Feasibility Analyses (Primary Outcomes): Feasibility analyses will focus on assessing economic and cultural acceptability, operability, and perceived/planned sustained use of the BUD app, including barriers and enhancing factors. From the corner store owner, wholesaler, and producer perspectives, acceptability will be assessed according to whether the new products are perceived to be in demand, profitable, and easy to obtain and store. Operability will be assessed primarily in terms of the BUD app, such as whether store owners indicate (and demonstrate) that they can and do continue to use the different features of the app. Perceived sustainability will be assessed in terms of whether store owners (and other users, such as wholesalers and producers) indicate that they plan to continue to use the BUD app, how much, and for what purposes. Textual data will be coded by trained staff using a codebook that emphasizes these feasibility constructs. Qualitative analysis will center on providing contextually rich descriptions of each aspect of feasibility and describing some of the key sources of variation (e.g., by store size, owner ethnicity, etc.). Analysis of the qualitative consumer interviews will focus on identifying those healthier products that would be most acceptable to them in terms of demand, cost, taste, and other similar characteristics. This analysis will also identify best in-store strategies to promote healthy foods to consumers.

Process Evaluation Analyses: We will conduct a detailed quantitative process evaluation (in terms of reach, dose delivered, and fidelity), and results will serve as additional measures of feasibility, particularly in terms of establishing operability and potential sustainability. Three primary process dimensions will be assessed: reach, dose delivered, and fidelity—each will likely have multiple measures—and standards will be set for each measure to monitor the quality of intervention implementation. We will set standards to monitor the quality of intervention implementation for the reach, dose, and fidelity measures, and revise these standards based on the experience of this pilot trial for the planned future full-scale trial.

Feasibility will also be assessed via analysis of user satisfaction responses collected during operation of the BUD app, as well as in terms of responses to the open-ended questions that will be part of the process evaluation measures.

Pilot Trial Store Impact Data Analyses (Secondary Outcomes): Using direct observation-based stocking data, an HFAI score will be calculated, and we will initially examine the difference from pre- to postintervention comparing intervention and control stores. We will also look at change in availability and sales of specific promoted product categories from pre- to postintervention at each specific phase of intervention. Mixed-effect models, treating the stores as a random effect, will be used to evaluate the effect of the intervention on the availability and sales of healthful food and beverage scores comparing corner store intervention and comparison groups and adjusting for covariates, such as store size and food stocking at baseline. We hypothesize that the prices of some promoted foods and beverages will decrease given that promoted foods/beverages will be subsidized during the first two phases of the intervention, as compared with baseline (if stocked at baseline). Furthermore, we will conduct additional tests to see whether a pass-through price effect (i.e., when the reduced costs to the retailer provided by a subsidy lead to a reduced price to the consumer) occurred related to the initial subsidy. Since the subsidy is short term (just phases 1 and 2 of the intervention), relatively small, and intended to generate initial usage of the app, we do not anticipate a significant price effect.

Pilot Trial Adult Consumer Impact Data Analyses: From the AIQ on the purchasing frequency of promoted foods and beverages (i.e., fruit and vegetables, low-fat snacks, low-sugar beverages, whole grain foods) and using the Block FFQ data on the intake of the same products, we will compare the change in purchasing and consumption among consumers of intervention stores with the comparison store consumers. These analyses will be adjusted for potential confounders, such as age, sex, income, participation in food assistance programs, and other household covariates, when appropriate. We will use linear mixed-effect models with individual and store random effects to identify potential predictors of participation. We hypothesize that there will be a trend toward increased purchasing and consumption of healthy promoted foods and beverages among intervention consumers compared with consumers that are sampled from comparison stores.

Economic Analyses: We will conduct economic analyses from three perspectives. First, we will study the economic impact of adding BUD from the store’s owner perspective. In this approach, we will use a marginal analysis approach where we will evaluate the sequential benefits of progressively adding the BUD app to cover a larger proportion of transactions versus the marginal costs of an incremental adoption. We will also look at the adoption from a one-time decision analysis where we will compare the total stream of benefits with the total costs of developing and adopting BUD. We will pay particular attention to fluctuations in profits and cost efficiency gains in terms of reduction in costs of repeated transactions and other potential operational gains. Second, we will consider an analysis at the community member level, where we will examine the benefits and costs to the consumers of introducing the app. In this case, we will consider reduction in prices, availability of healthy items, and if possible, changes in the quality of products. Finally, we plan to study how psychological constraints may influence the individual judgment of adopting the technology (framing of the photos of the food, simplicity of use, size of the consequences of errors, social norms, fear to change previous practices, etc.). This broader approach to studying the economic implications of adopting the technology is important for understanding its potential sustained use.

Upon completion of data collection, entry, and cleaning, the data analysis team will meet to discuss missingness and build an appropriate model for imputation, including identification of variables for imputation (i.e., missing data that fit the assumptions of missing at random (MAR)) and auxiliary variables that should be included in the imputation model. We will use a method known as fully conditional specification (FCS), also referred to as multiple imputation by chained equations (MICE), which allows for imputation on a variable-by-variable basis, cycling iteratively through univariate models for each variable conditioned on all other variables in the model.

Criteria for Success: We have established criteria for success of this feasibility trial, which will support applying for an NIH R01m, which will be a full-scale implementation of the BUD app in multiple cities, and which will be powered to show impact at the consumer level. These criteria include: evidence of high acceptability of the app as reported by small store owners and their suppliers, strong operability as provided by usage data, and high satisfaction with the app. In addition, the criteria for success will include significant increases in the stocking and sales of promoted foods in corner stores. We will also examine changes in consumer diet and hope to show preliminary efficacy. Finally, we will assess the recruitment and retention of suppliers, retailers, and consumers in order support a planned full-scale trial.

## 3. Discussion

We describe the development and piloting plan for assessing the feasibility of the Baltimore Urban food Distribution (BUD) app, a novel food systems intervention to improve the distribution of food to small urban corner stores from local producers and wholesalers. There are other interventions aimed at improving the food system (sugar-sweetened beverage (SSB) taxes, Baltimore Healthy Stores), but almost all of these focus on just one level of the food system (e.g., policy, food stores) and employ a single strategy (e.g., raising the prices of SSB) [30,31]. Shape Up Somerville and B’More Healthy Communities for Kids are two previously implemented interventions that sought to impact multiple levels of the food system simultaneously, but did not focus heavily on influencing the interaction between levels [32,33]. BUD directly impacts multiple levels simultaneously (retailers, suppliers, and consumers), and does so by building networks between retailers, linking them to suppliers, integrating different types of suppliers, and employing multiple supportive strategies (collective purchasing, shared delivery). The app is expected to stimulate collaboration between small store owners and lead to enhanced communications between store owners and their suppliers.

The BUD intervention is designed to target the specific food access needs and constraints of low-income urban communities. It is based on our nearly 20 years of experience working with local retailers to improve the Baltimore food system [29,34,35,36,37,38,39]. Many challenges exist that we hope to address in this pilot trial. The first is one of acceptability. Will small store owners use the app, and what features will encourage their use? Store owners’ gains for adopting the BUD app must be greater than costs, and these net benefits must be recurrent for individuals to adopt the new technology over a sustained period. Personalized training in the use of the BUD app may be helpful, but will be difficult to sustain if the use of the BUD app expands. We will get an early idea of sustainability during phases 3 and 4 of implementing the app, when no subsidy is provided. The BUD app uses two features to reduce costs: collective purchasing and shared delivery. However, the degree to which store owners will use these features is uncertain. A related consideration is financial sustainability and expansion. BUD could potentially become self-supporting. We anticipate that after the initial subsidy is completed, some stores will see the benefit of the app in terms of convenience and costs and will begin to use the app on their own.

Another challenge will be how to make the BUD app responsive to public health priorities and to focus on supplying mainly healthier foods—we do not want BUD to become another way to get chips and candy into urban corner stores. This is a real concern. In our early formative work [23], corner store owners were clear that they would prefer that the app allowed them to access a wide variety of products and not just a limited number of healthier products. Following this pilot, we plan to test versions of the app that promote healthier items, through, for example, app-subsidized pricing and/or nudging strategies. Other plans include designing and developing a consumer-engaged module of the BUD app, called “BUDConnect”, which will support communications between small store owners and the consumers they serve.

In regard to data analysis, there is a potential for cohort effects to bias our findings, although we do not feel that this limitation will significantly impact our study given its longitudinal design due to the fact we have comparison groups at the store and consumer levels.

Our other future plans, if this pilot meets our criteria for success, include a full-scale implementation in multiple cities (i.e., Philadelphia, Washington DC, Detroit, etc.) where there are noted gaps in food distribution with no digital multilevel interventions in place. There are multiple challenges in “scaling up”, including the need to develop similar relationships in new places, evaluating assets and needs in new settings, making adaptations, and identifying resources. However, it is highly likely that some interventions, including the current one, may be adaptable to a variety of different urban areas, particularly those urban areas that share several similar assets and challenges. Adapting a food system intervention to other urban food environments could save time and resources.

A decentralized approach to scale up the BUD app could also be pursued, in which local public health researchers, working with software development teams, can develop local versions of the BUD app, tailored according to the needs of their communities. To facilitate this scale-up, we will complement our dissemination of the BUD project protocols and findings with the publication of the actual software components that drive the BUD app. As noted earlier, this can be performed through the widely used GitHub software repository along with online tutorials that will provide software development teams with the technical details, including step-by-step implementation procedures, that are needed to make BUD operational. We will also highlight the BUD back-end and front-end components that can be modified to suit the needs and requirements of local users.

## 4. Conclusions

In summary, the BUD app is a food-system-changing intervention that is in development and will be piloted in 2022–2023. We anticipate that the app will provide a financially viable alternative for small food stores in Baltimore to increase their stocking and sales of healthier foods, subsequently increasing healthy food access and potentially improving diet-related health outcomes for frequent customers of these stores who are otherwise under-resourced. After the pilot is completed, we will seek funding to refine and scale up the app to urban and rural settings across the United States and beyond.

## Figures and Tables

**Figure 1 ijerph-19-09138-f001:**
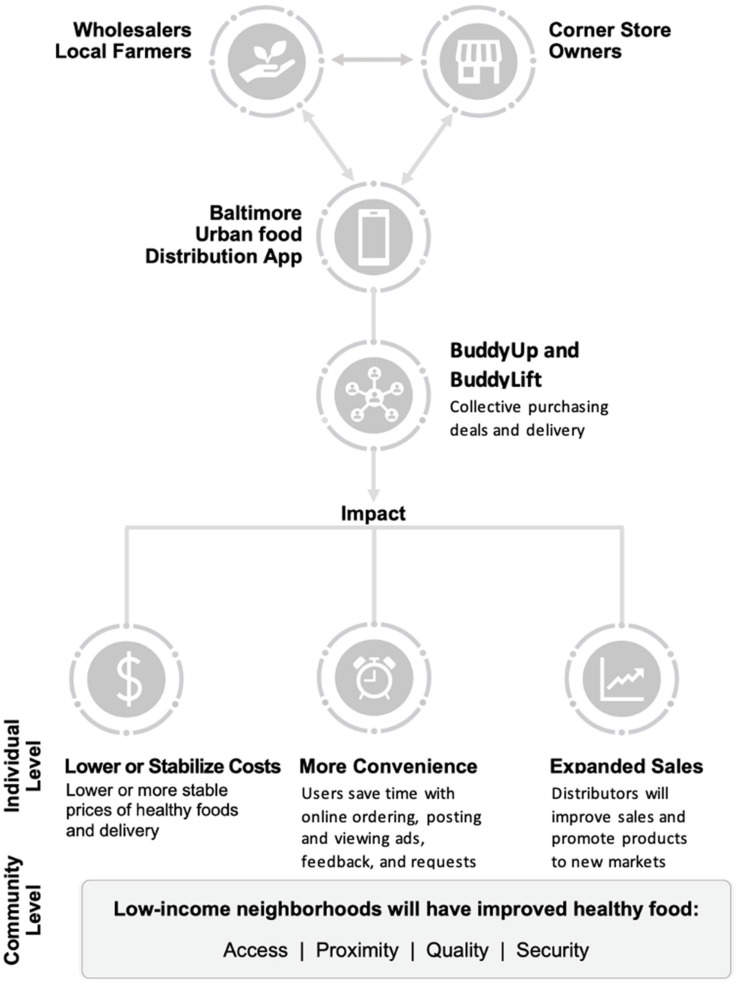
Baltimore Urban food Distribution (BUD) app conceptual framework.

**Figure 2 ijerph-19-09138-f002:**
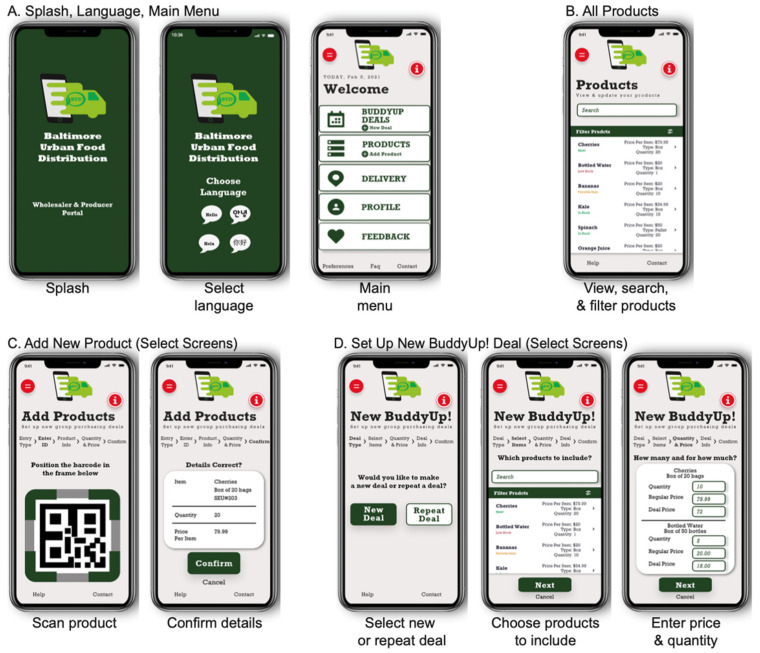
BUD producer and wholesaler screen mockups.

**Figure 3 ijerph-19-09138-f003:**
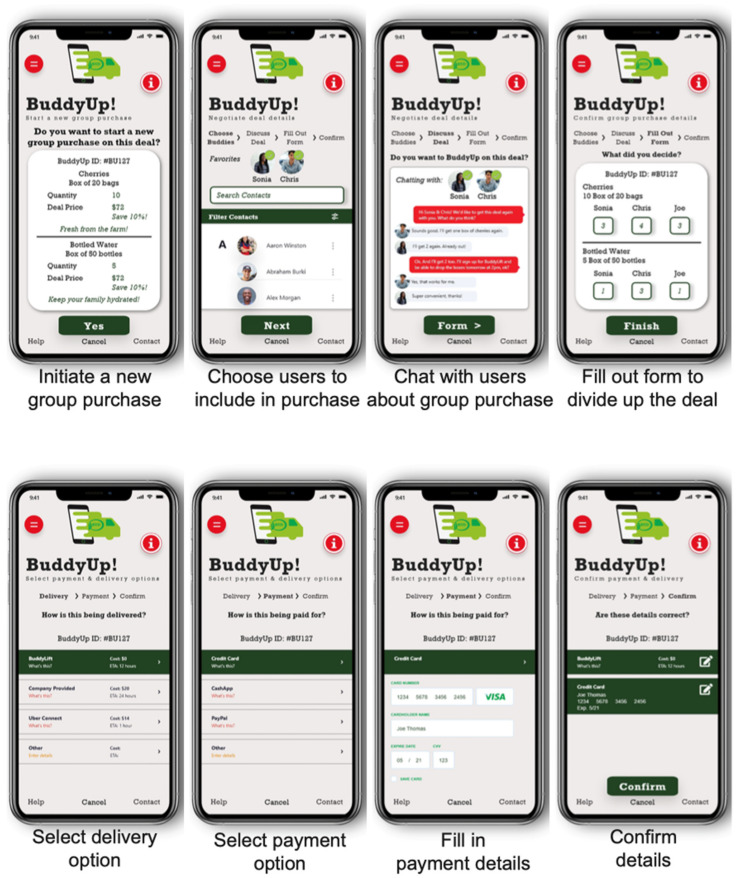
BUD corner store owner screen mockups.

**Table 1 ijerph-19-09138-t001:** Stages of implementation of the BUD app during the pilot.

BUD Strategies	Stage 1	Stage 2	Stage 3	Stage 4
App Features	BUD App	BUD App + BuddyUp!	BUD App + BuddyUp! + BuddyLift!	BUD App + BuddyUp! + BuddyLift!
Promoted Foods	Low-Sugar Beverages	Fresh Fruits and Vegetables	Low-Fat Whole Grains, Snacks	Low-Sugar Beverages + FV + Whole Grains, Snacks
BUDCredit (for corner stores)	$100	$100	$0	$0

**Table 2 ijerph-19-09138-t002:** Summary of impact and process measures by level of data collection.

Type of Data Collection	Time of Data Collection	Level of Data Collection
Feasibility metrics: acceptability, operability, perceived sustainability and user satisfaction with the BUD app	During intervention (multiple measures), Post-intervention	Producer, Wholesaler, Retailer
Stocking of healthy and unhealthy foods	Baseline, During intervention, Post-intervention	Retailer
Process metrics: reach, dose delivered, fidelity	During intervention (multiple measures)	Study Team
Sales of healthy and unhealthy foods	Baseline, During intervention, Post-intervention	Retailer
Purchasing of healthy foods	Baseline, Post-intervention	Consumer
Consumption of healthy foods	Baseline, Post-intervention	Consumer
Financial costs and benefits from perspective of suppliers	Post-intervention	Producer, Wholesaler, Supplier
Prices of healthy foods	Baseline, During intervention, Post-intervention	Retailer

## Data Availability

De-identified study data will be made available following completion of the trial in consultation with the study PI.

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
