# Peer review of "The Baltimore Urban Food Distribution (BUD) App: Study Protocol to Assess the Feasibility of a Food Systems Intervention"

_ijerph, 2022, doi:10.3390/ijerph19159138_

Round 1
Reviewer 1 Report
The manuscript investigates on a relevant topic for the U.S. food market, such as the food access as it impacts on consumers' health. The manuscript is well-written and organized, as well as the case study analysis is properly conducted, and it provides interesting results. However, I strongly encourage the authors to add a literature review in the introduction on studies investigating food access in the U.S. See for example:
Bonanno, A., & Li, J. (2015). Food insecurity and food access in US metropolitan areas. Applied Economic Perspectives and Policy, 37(2), 177-204.
Taylor, R., & Villas-Boas, S. B. (2016). Food store choices of poor households: A discrete choice analysis of the National Household Food Acquisition and Purchase Survey (FoodAPS). American Journal of Agricultural Economics, 98(2), 513-532.
Courtemanche, C., Carden, A., Zhou, X., & Ndirangu, M. (2019). Do Walmart supercenters improve food security?. Applied Economic Perspectives and Policy, 41(2), 177-198.
Reviewer 2 Report
This is a protocol study discussing to develop a digital application (app) to improve access to healthier foods and beverages in small corner stores in low-income urban settings, and to generate preliminary findings in support of a full-scale clinical trial. I think the topic is interesting and contributive to the food science under a systematic protocol approach for food science professionals.
Minor
1. Please add statistical analysis section after a systematic review protocol. I mean which statistical methods used in this study? T test? Chi-square test? The authors should clarify this concern.
2. In my opinions, research questions should be divided primary and secondary objective. The authors should clarify this concern.
3. Which system used for electronic data capture? The authors should clarify this concern.
4. Although the role of novel food systems is clear, cohort effect also become a bias existed in both genders for this topic besides food systems. The authors should clarify this concern.
5. It is not clear whether food systems affect lived experiences or diseases, and it is important to obtain conclusions from the system review and meta-analysis. Please boldly draw the expected conclusion form this protocol study for public health perspective.
6. Most of the discussion is introductory, probably because there are no results to discuss. Please added some solution for the expected conclusion form this protocol study.
Reviewer 3 Report
General:
This was an interesting protocol paper to read to address what is a global challenge. Most of my comments are minor and some optional. I do feel the more wider reaching use of this protocol beyond that of the current setting in which the study is being undertaken needs to be more pronounced throughout (see comments). Furthermore, there were a number of underdeveloped claims that need to be further discussed.
Abstract:
Subtitling is fine (quite useful in fact) but the discussion sub-sections adds nothing useful or insightful. There are several interesting comments made in the discussion. Please revise with a clear statement of the contribution that the study (or the current protocol paper) will bring.
Introduction:
Generally, this section is well written with a good level of flow between concepts. That said, I don’t think the utility of the app and its purpose (what it should / can do) has been fully developed. There is no real context as to how an app would do this and how it begins to address the challenges highlighted. This needs to be reviewed more thoroughly. I understand that further context is provided in the method, but as a reader I was left without the full picture before continuing.
Other minor comments –
L43-45 appears to be a repetition of the sentences previous? Could I suggest these are combine or omitted as appropriate.
L47 – could the authors clarify what they mean by “protective” in this sense.
L58 – I don’t see the utility of a rhetorical question here. This is personal preference so feel free to ignore, but it doesn’t add anything.
Optional – may I suggest the authors place L 68-73 before the statement of L64-67 above as it adds a bit more context to what the authors are referring to by multi-level food interventions (and the interacting nature)?
Method:
L114-118 repeats information of the introduction section and needs a solid example of the extent of not stocking healthy foods (this in itself is ambiguous). Are there any clear stats / percentages on foods stuffs that can be communicated here?
L118 – Is the term “white neighbourhoods” necessary? Would middle-high income areas suffice?
Moreover, could the comparison claim made here be reconceptualised with how many supermarkets are in such middle-high income areas (or that there is an improved range to these), which would help the reader understand a little more the information indicated at the start of the paragraph.
L148 – ethics approval number needed here and or at the relevant section at the end of the manuscript.
Regarding the pre-pilot will any stores that took part be ineligible for the main intervention?
As per one of my comments above, some of the details of how the app is enabling improved access to food access for both customer and consumer need to be communicated in the introduction section. Some of these ideas are very interesting but there is no context provided until page 9-10. This is a real shame and actually makes it quite hard to follow what the intervention is until this point. I would strongly recommend to address this.
The analysis approach is well documented but could this be better organised?
Discussion:
It is great to see the authors highlighting potential issues and learning points at this stage, although please consider moving away from excessive rhetorical questions. More suggestions of how the team are considering to deal with such would be more useful.
It would be useful to really stress the implications of this work and its application to other areas (US wide / or hopefully even beyond). Stating that the team will look at other cities just feels very vague.
More linking back to why an app is the best approach is again needed.
The suggestion to enhance via nudging is a good idea but this would naturally be contextualised by the engagement to start with. How would the authors propose keeping the users engaged after the incentives have finished?
Reviewer 4 Report
I commend the authors for the development of a research protocol titled, "t] "The Baltimore Urban food Distribution (BUD) App. Study protocol to assess the feasibility of a food system intervention."
1. Was the development of the BUD app based on a community needs assessment conducted in the study area?
2. There are three main mobile operating systems – iOS, Android, and Windows – and in order to achieve the best out of the app, it needs to be available on all of these. Which of these operating systems will the BUD App be operating on?
3. Internet security is becoming an ever-increasing issue and the same applies to your app. With many applications storing personal and sensitive information or credit and debit card details, security is an absolute must. What security provisions, if any, did you include in your protocol?
4. Why is the BUD App limited to food distribution? Why not include other essential grocery supplies?
5. What continuous improvements and update plans, if any are included in your BUD App?
Round 2
Reviewer 1 Report
I am fine with the current vesion of the manuscript